# Sweet Pepper (*Capsicum annuum* L.) Fruits Contain an Atypical Peroxisomal Catalase That Is Modulated by Reactive Oxygen and Nitrogen Species

**DOI:** 10.3390/antiox8090374

**Published:** 2019-09-04

**Authors:** Marta Rodríguez-Ruiz, Salvador González-Gordo, Amanda Cañas, María Jesús Campos, Alberto Paradela, Francisco J. Corpas, José M. Palma

**Affiliations:** 1Group Antioxidant, Free Radicals and Nitric Oxide in Biotechnology, Food and Agriculture, Department of Biochemistry, Cell and Molecular Biology of Plants, Estación Experimental del Zaidín, CSIC, 18008 Granada, Spain (M.R.-R.) (S.G.-G.) (A.C.) (M.J.C.) (F.J.C.); 2Proteomics Core Facility, Centro Nacional de Biotecnología, CSIC, 28049 Madrid, Spain

**Keywords:** bovine, catalase, molecular characterization, nLC-MS/MS, pepper fruit ripening, peroxisome, quaternary structure, reactive oxygen species, reactive nitrogen species, *S*-nitrosation

## Abstract

During the ripening of sweet pepper (*Capsicum annuum* L.) fruits, in a genetically controlled scenario, enormous metabolic changes occur that affect the physiology of most cell compartments. Peroxisomal catalase gene expression decreases after pepper fruit ripening, while the enzyme is also susceptible to undergo post-translational modifications (nitration, *S*-nitrosation, and oxidation) promoted by reactive oxygen and nitrogen species (ROS/RNS). Unlike most plant catalases, the pepper fruit enzyme acts as a homodimer, with an atypical native molecular mass of 125 to 135 kDa and an isoelectric point of 7.4, which is higher than that of most plant catalases. These data suggest that ROS/RNS could be essential to modulate the role of catalase in maintaining basic cellular peroxisomal functions during pepper fruit ripening when nitro-oxidative stress occurs. Using catalase from bovine liver as a model and biotin-switch labeling, in-gel trypsin digestion, and nanoliquid chromatography coupled with mass spectrometry, it was found that Cys377 from the bovine enzyme could potentially undergo *S*-nitrosation. To our knowledge, this is the first report of a cysteine residue from catalase that can be post-translationally modified by *S*-nitrosation, which makes it especially important to find the target points where the enzyme can be modulated under either physiological or adverse conditions.

## 1. Introduction

Catalase, which has been thoroughly reported to be located in peroxisomes, is one of the principal antioxidant enzymes in plants [1,2,3,4,5,6]. This iron-containing homotretameric heme protein is involved in the dismutation of H_2_O_2_ into H_2_O and O_2_ [6,7,8]. The native plant enzyme ranges in size from 220 to 240 kDa and is composed of four identical subunits of approximately 55 to 59 kDa [1,3,4,8,9,10,11,12]. Interestingly, the catalase enzyme, isolated from leaves of the halophyte *Mesembryanthemun crystallinum*, with a molecular mass of approximately 320 kDa, can be divided into two less kinetically active dimeric forms of catalase, each measuring 160 kDa [13]. Catalase, which plays an important role in plant growth, development, and stress responses [3,4,6,14,15], is also associated with fruit ripening and postharvest events [16,17,18,19,20,21].

Catalases have a highly diverse range of structures and molecular properties, with hundreds of gene and protein sequences being reported [3,22,23]. Unlike animals, which contain a single catalase gene, in plants, this enzyme is encoded by a multigene family, which provides multiple isozymes whose number varies depending on the species (see [6] and references therein). For example, using gene analysis, three classes of catalase isozyme genes have been identified in maize (*Zea mays*) and rice (*Oryza sativa*), whose expression is regulated according to their distribution and the environmental conditions [24,25]. Like maize and rice, the genome of the model plant *Arabidopsis thaliana* contains three catalase genes [26,27,28,29], while the cotton (*Gossypium* ssp.) genome is composed of up to seven genes [15], with the barley (*Hordeum vulgare*) [30] and peach (*Prunus persica*) [11] genomes each containing two catalase genes. Using biochemical techniques, a variable number of catalase isozymes have also been reported, with four isozymes identified in loblolly pine (*Pinus taeda*) megagametophytes [9], five in pea (*Pisum sativum*) leaves [1], and one in lentil (*Lens culinaris*) leaves [31].

As indicated above, this variability is explained by the differential expression patterns of each class of gene, as determined by the plant developmental stage, target tissue/organ, and environmental conditions [2,3,6]. However, we still have a way to go in understanding the tight regulation of this enzyme, which can be mediated by signaling molecules. Accumulating evidence in recent years suggests that the antioxidative metabolism, in which catalase plays a prominent role, is regulated via signaling events promoted by nitric oxide (NO) and its derived molecules called reactive nitrogen species (RNS) [32,33,34]. In pepper fruits, catalase activity decreases during ripening [35], which could be due to tyrosine nitration, a post-translational modification (PMT) caused by reaction with peroynitrite (ONOO^−^), an RNS formed by the reaction of nitric oxide (NO) with superoxide radicals (O_2_^•−^) [18,36,37].

Pepper (*Capsicum annuum* L.), which is mainly characterized by its high vitamin C and A content, is one of the most consumed vegetables in the world [20,38]. During fruit ripening, chlorophyll breakdown and synthesis of new carotenoids and anthocyanins take place in an intense metabolism characterized by the emission of volatile organic compounds associated with respiration, change to acidic taste, pH and astringency changes, pectin formation, new protein synthesis, cleavage of existing proteins, conversion of starch to simple sugars, flavor accumulation, and cell wall softening, among other events [20,39,40,41,42,43]. The reactive oxygen species (ROS) metabolism is also affected during this physiological process, which leads to major changes in total soluble reducing equivalents in fruits [20,40,44]. At the subcellular level, in the peroxisomal compartment where catalase is located in plants, important metabolic changes involving the ROS metabolism occur during pepper fruit ripening [35]. Recently, using proteomic techniques, catalase was found to be the most commonly identified peroxisomal protein [20,21].

Being one of the most important and abundant antioxidant plant enzymes and given its potential key role during fruit ripening, this study provides a biochemical characterization of catalase in pepper fruits and its modulation by NO-derived molecules and reducing agents.

## 2. Materials and Methods

### 2.1. Plant Material

California-type sweet pepper (*Capsicum annuum* L.) fruits were obtained from plants grown in plastic-covered greenhouses (Syngenta Seeds, Ltd., El Ejido, Almería, Spain). Immature, green-stage fruits, which display higher activity than ripe fruits according to previous data obtained in our laboratory, were used in our analyses [35,45]. After harvesting, the fruits were cut into small cubes (approximately 5 mm/side), frozen under liquid nitrogen, and then stored at −80 °C until use.

### 2.2. Preparation of Crude Extracts for Enzyme Activity

Frozen samples of sweet pepper fruits were powdered under liquid nitrogen using an IKA^®^ A11 Basic analytical mill (IKA^®^, Staufen, Germany) and then extracted in 0.1 M Tris-HCl buffer, pH 8.0, containing 1 mM ethylenediaminetetraacetic acid (EDTA), 0.1% (*v*/*v*) Triton X-100, 10% (*v*/*v*) glycerol in a final 1:1 (*w*:*v*) plant material:buffer ratio. Homogenates were centrifuged at 15,000× *g* for 30 min and the supernatants were used for enzymatic assays.

### 2.3. Catalase Activity Assays: Spectrophotometry and Non-Denaturing Electrophoresis—Effect of Different Modulators

Catalase (EC 1.11.1.6) activity was determined by monitoring the decomposition of H_2_O_2_ at 240 nm as described by Aebi [46]. Protein concentrations in the samples were monitored using a Bio-Rad protein assay solution (Bio-Rad Laboratories, Hercules, CA, USA), with bovine serum albumin as standard [47].

To analyze the catalase isoenzymes, nondenaturing polyacrylamide gel electrophoresis (PAGE) and isoelectric focusing (IEF) were performed (see below), and gel activity was visualized according to the method described by Clare et al. [48]. Briefly, after the electrophoretic procedures were carried out, the gels were incubated for 45 min in a solution containing 50 µg/mL horseradish peroxidase (Type II, 250 units/mg, Merck KGaA, Darmstadt, Germany), prepared in 50 mM phosphate buffer, pH 7. Then, 0.5 mM H_2_O_2_ was added to the solution followed by further incubation for 10 min in the dark. Finally, the gels were stained with 0.5 mg/mL 3,3-diaminobenzidine (DAB) in phosphate buffer until colorless bands appeared over a brown background.

To analyze the modulation in enzyme activity in in vitro assays, pepper fruit samples were incubated at 25 °C for 45 min with different modulation agents, including 3-morpholinosydnonimine (SIN-1, 2 mM), a peroxynitrite (ONOO^−^) donor, diethylamine NONOate (DEA-NONOate, 2 mM), and *S*-nitrosoglutathione (GSNO, 0–6 mM) as NO donors, and reduced glutathione (GSH, 0–6 mM) and dithiotreitol (DTT, 5 mM) as reducing compounds [18,49]. In the assays, incubations for 30 min in the presence of different concentrations of the chemical H_2_O_2_ (0–50 mM) as oxidant were also performed. In all cases, the solutions were made up fresh before use.

Additionally, for more specific studies of catalase *S*-nitrosation (see below), the enzyme from bovine liver (aqueous suspension, 10,000–40,000 units/mg protein, Sigma-Aldrich) was incubated with different concentrations (0–1 mM) of either *S*-nitroso-L-cysteine (CSNO) or L-cysteine for 1 h at 25 °C in the dark, and enzyme activity was then determined.

### 2.4. Electrophoretic Methods

Nondenaturing PAGE was carried out at 5% to 7% acrylamide concentrations in a Mini-Protean Tetra Cell (Bio-Rad, Hercules, CA, USA). Next, 1.5 mm-thick gels were prepared in 375 mM Tris-HCl, pH 8.9, in a 19:1 acrylamide:bis-acrilamide ratio. Pepper fruit samples were added 0.006% (*w*/*v*) bromophenol blue dye and then loaded onto gels. Pre-electrophoresis was initially run at 15 mA/gel for 30 min and then at 25 mA/gel until the dye front reached 1 cm above the gel edge.

As mentioned above, IEF was achieved [50]. Gels (1.5-mm thick) contained 6% acrylamide, 2.3% (*w*/*v*) ampholytes, pH 3.5–7 (General Electric Healthcare Life Sciences, Chicago, Illinois, USA), and 10% (*v*/*v*) glycerol. Samples were prepared with 15% (*w*/*v*) sucrose and 2.3% (*w*/*v*) ampholytes, pH 3.5–7. Once the samples were deposited on the wells, a protective layer of 7.5% (*w*/*v*) sucrose and 2.3% (*w*/*v*) ampholytes, pH 3.5–7, was loaded on top. A Mini-Protean Tetra Cell (Bio-Rad, Hercules, CA, USA) system was set up for the assay, using 0.1 M NaOH (cathode) and 0.06% (*v*/*v*) H_3_PO_4_ (anode) as electrode solutions. The electrophoretic procedure at 4 °C and constant voltages was carried out in the following sequence: 150 V, for 30 min; 200 V for 30 min; and 250 V for 1.5 to 2 h. IEF standards (Bio-Rad, Hercules, CA, USA) were used as isoelectric point (pI) markers.

PAGE under denaturing conditions in the presence of sodium dodecyl sulfate (SDS-PAGE) was also performed [51]. Gels (12% acrylamide) were prepared in 375 mM Tris-HCl, pH 8.9, 10% (*v*/*v*) glycerol, and 0.1% (*w*/*v*) SDS, and samples were run in a Mini-Protean Tetra Cell (Bio-Rad, Hercules, CA, USA) at 200 V until the dye front reached 1 cm above the gel edge. Prestained Precision Plus Protein Dual Color Standards (Bio-Rad, Hercules, CA, USA) were used as molecular size markers.

Proteins were stained in gels using either Bio-Safe^TM^ Coomassie Stain (Bio-Rad, Hercules, CA, USA) or, when necessary, silver staining according to the method described by Heukeshoven and Dernick [52].

### 2.5. Immunoblot Analysis

Proteins separated by either SDS- or non-denaturing PAGE were transferred onto polyvinylidene difluoride (PVDF) membranes using Trans-Blot SD (Bio-Rad, Hercules, CA, USA) and 10 mM N-cyclohexyl-3-aminopropanesulfonic acid (CAPS) buffer, pH 11.0, containing 10% (*v*/*v*) methanol as transfer buffer. The procedure was run at 1.5 mA/cm^2^ membrane for 2 h [53]. After the electrophoretic run, membranes were processed for a further blotting assay. A primary antibody against catalase from *Arabidopsis thaliana* (Agrisera AB, Vännäs, Sweden) (dilution 1:5000), and goat anti-rabbit conjugated to horseradish peroxidase (dilution 1:20,000, Bio-Rad, Hercules, CA, USA) as secondary antibody were used. The antibody-recognizing proteins were detected using the Clarity™ Western ECL Substrate kit (Bio-Rad, Hercules, CA, USA) according to the manufacturer’s instructions.

### 2.6. Determination of Native Molecular Weight of Catalase

Two techniques were used to determine the native molecular weight of catalase from pepper fruits: Non-denaturing PAGE at different acrylamide concentrations and gel filtration through a chromatographic procedure.

The Hedrick and Smith method was used for the electrophoretic analysis [54]. This involves a series of non-denaturing electrophoreses at different acrylamide concentrations (5–7%), with sample preparation and running conditions as indicated above. Pepper samples and standard proteins with known molecular weights were loaded onto the same gels. Monomer, dimer, trimer, and tetramer forms of BSA were used as molecular weight standards, and commercial bovine catalase (Sigma-Aldrich, Darmstadt, Germany) as the positive activity control. After electrophoresis, standard proteins were stained with Bio-SafeTM Coomassie (Bio-Rad, Hercules, CA, USA), while catalase activity in pepper samples was detected in gels as described above. Taking into account the relative electrophoretic mobility (Rf) of standards and catalase bands in gels, 100 log (Rf × 100) was plotted against acrylamide concentrations. The standards were then plotted against molecular weight. The slopes for catalase from pepper samples were interpolated in the curve built using the standards plot to calculate the theoretical weight values.

For the gel filtration technique, fast protein liquid chromatography (FPLC; Äkta, General Electric Healthcare Life Sciences, Chicago, Illinois, USA) equipment was used. Crude extracts from green pepper fruits were prepared as indicated above and were then filtered through two nylon cloth layers. After centrifugation at 15,000× *g* for 30 min, samples were concentrated by fractionated precipitation with (NH_4_)_2_SO_4_ (20–80%). The final pellet, containing the bulk of catalase activity, was re-suspended in 50 mM Tris-HCl, pH 8.0, and centrifuged as described above. The supernatant was then cleaned using a PD-10 desalting column containing Sephadex™ G-25 (General Electric Healthcare Life Sciences) before being used for the FPLC assay. Samples (200 µL) were loaded onto a Superose 12 column (General Electric Healthcare Life Sciences), and the elution was run with the reported buffer. In the collected fractions (500 µL), absorbance at 280 nm was recorded, and catalase activity [46] was determined. The fractions were also analyzed by SDS-PAGE and western blotting using an antibody against catalase from *A. thaliana*. To calibrate the Superose 12 column, the following standard proteins were used: Rubisco (Mw = 560,000 Da), ferritin (Mw = 450,000 Da), aldolase (Mw = 160,000 Da), phosphorylase b (Mw = 97,600 Da), and myoglobin (Mw = 18,700 Da). A solution containing all standard proteins was prepared in 50 mM Tris-HCl, pH 8.0, each at a 1 mg/mL concentration, and 200 µL was loaded onto the column. In the eluted fractions, absorbance at 280 nm was read, and log Mw was plotted against the elution volume. The elution volume of fractions containing catalase activity was used to interpolate the standard curve to obtain the enzyme’s native molecular weight.

### 2.7. Bovine Catalase as a Model to Analyze the S-Nitrosation Process

In these assays, commercial catalase from bovine liver (aqueous suspension, 10,000–40,000 units/mg protein, Sigma-Aldrich, Darmstadt, Germany) and the biotin-switch labeling technique were used [55]. Two aliquots of commercial catalase (1 µg/µL) were prepared. One aliquot was incubated with 1 mM S-nitroso-L-cysteine (CSNO) for 1 h at room temperature in darkness to generate *S*-nitrosation of susceptible thiol groups. The other aliquot was incubated with 5 mM N-ethylmaeimide (NEM) in the same conditions to block all free thiols (used as internal control), and afterwards both catalase aliquots were used for the standard biotin-switch labeling technique. Then, methyl methanethiosulfonate (MMTS) was then added up to a final concentration of 20 mM and incubated at 50 °C for 1 h with continuous shaking. Two acetone precipitation steps were performed. Acetone at −20 °C was added in a 2:1 acetone:sample ratio and was maintained at this temperature for 1 h. After centrifugation at 15,000× *g* for 10 min, the supernatant was discarded, and the pellet was left to dry at room temperature. The pellet was resuspended in HENS buffer [100 mM 2-[4-(2-hydroxyethyl)piperazin-1-yl]ethanesulfonic acid (HEPES), pH 7.8, 1 mM EDTA, 0.1 mM neocuproine, and 1% (*w*/*v*) SDS]. The acetone procedure was then carried out one more time. The final pellet was resuspended again in HENS buffer with 1.25 mM ascorbate and 4 mM N-[6-(biotinamido)hexyl]-3′-(2′-pyridyldithio)propionamide (biotin-HPDP) and incubated in the dark for 3 h with continuous shaking. Acetone (–20 °C) was then added in a 2:1 acetone:sample ratio and centrifuged at 15,000× *g* for 10 min. The pellet was resuspended with electrophoresis loading buffer (without reducing agent) and subjected to SDS-PAGE in 12% acrylamide gels.

Polypeptides were stained in gels with Bio-SafeTM Coomassie staining solution (Bio-Rad). The catalase stained bands (control treated with NEM, and that treated with CSNO) were excised from the gel and digested overnight with trypsin under reducing conditions. The digested peptides were recovered, dried in a speed-vac, and analyzed using reversed-phase nano liquid chromatography (nLC) with a C18 column and a short gradient (40 min). This system was coupled to a SCiex 5600 Triple-Q-TOF mass spectrometer (AB Sciex, Old Connecticut Path Framingham, Massachusetts, USA) in data dependent acquisition (DDA) mode. Tandem mass spectrometry (MSMS) spectra were used to search against the reference *Bos taurus* proteome (https://www.uniprot.org/uniprot/?query=proteome: UP000009136). Standard search criteria (mass tolerance, missed cleavages) were used, and the variable modifications considered were protein N-terminal acetylation, methionine oxidation, and cysteine modification using either MMTS- or HPDP-biotin.

### 2.8. Statistical Analysis

With the aid of the Statgraphics Centurion program, analysis of variance (ANOVA) and the *t*-student test were used to detect differences in treatments. Values for *p* < 0.05 were considered statistically significant.

## 3. Results

### 3.1. Pepper Fruit Catalase Activity under Different Assay Conditions

During pepper fruit ripening, catalase, one of the most abundant peroxisomal enzymes associated with the antioxidant/oxidative metabolism, is among the down-expressed proteins, with higher gene expression, protein, and activity levels being observed in immature green fruits than in ripe red fruits [56]. In addition, this enzyme from pepper fruits has recently been shown to be inhibited by the post-translational modification (PTM) tyrosine nitration [18]. In light of these important modulation features, we further characterized this protein in immature green pepper fruits, with higher catalase activity than in red fruits. Initially, catalase activity in pepper fruits, which remained constant after crude extracts of pepper fruits were incubated at 4 °C for 3 days, showed considerable stability (Figure 1a). This finding was confirmed by an assay of catalase activity in non-denaturing gels, in which no isoenzymatic changes were observed after 72 h. As indicated in Figure 1b (activity staining), only one catalase isoenzyme band was detected in gels in all cases. This same isoenzyme profile, showing only one activity band, was also observed in ripe red fruits (results not shown). Specific catalase protein content was also investigated with the aid of non-denaturing PAGE followed by immunoblotting using an antibody against plant catalase. Once again, only one immunoreactive band was detected, with no changes in intensity being observed after 72 h of incubation at 4 °C (Figure 1b). Previous studies carried out by our group, using immunoblotting after SDS-PAGE analysis, showed that catalase from pepper fruits is characterized by single monomeric band (56 kDa) dissociation [18]. The presence of just one catalase isoenzymatic band was corroborated using the isoelectric focusing method, which detected only one activity band with an isoelectric point of 7.4 in gels (results not shown).

We also analyzed catalase enzyme activity after the pepper crude extracts were incubated with increasing concentrations of H_2_O_2_. Although catalase commonly reacts to hydrogen peroxide substrates, further analysis of enzyme kinetics showed that the incubation of extracts with more than 5 mM H_2_O_2_ for 30 min at 25 °C inhibited catalase activity by up to 40% at an H_2_O_2_ concentration of 50 mM (Figure 2a). This reduction in activity was confirmed by non-denaturing PAGE, which showed fainter activity bands as concentrations of H_2_O_2_ were increased during incubation (Figure 2b). These assays, which also detected only one activity band, showed higher electrophoretic mobility levels in 6% acrylamide gels with increasing concentrations of H_2_O_2_, thus suggesting potential catalase oxidation.

The effect of different modulators on the RNS metabolism was assayed by incubating pepper samples in the presence of these modulators before the catalase activity assay was carried out. This also confirmed the inhibitory effect of SIN-1, a tyrosine nitration-promoting peroxynitrite generator [18], while the potential *S*-nitrosation of catalase was also studied using the NO donors, DEA-NONOate and *S*-nitrosoglutathione (GSNO). As shown in Figure 3a, both NO donors inhibited catalase activity and, in the case of GSNO, in a concentration-dependent manner (Figure 3b). To determine whether this inhibition was due to the NO or glutathione (GSH) counterpart, catalase was also assayed in the presence of this reduced antioxidant. Our results indicate that catalase activity was down-regulated not only by *S*-nitrosation events but also by GSH (Figure 3b), whose effect was again concentration dependent.

### 3.2. Quaternary Structure of Pepper Fruit Catalase

In order to gain a deeper insight into how these modulations occur in pepper fruit catalase, a preliminary molecular characterization of the enzyme was carried out using catalase-enriched fractions. Thus, crude extracts from pepper fruits were fractionated by ammonium sulfate precipitation, which were then studied using FPLC size exclusion chromatography (see materials and methods above). Native molecular weight and structure were investigated using two distinct methods. The Hedrick and Smith electrophoretic method found that the native enzyme in non-denaturing gels [54] had a molecular weight of 135 kDa (Figure 4a), while the weight of the native enzyme determined by the gel filtration method was roughly similar, at approximately 125 kDa (Figure 4b).

### 3.3. Estimation of the S-Nitrosated Cysteine from the Bovine Catalase Used a Model

Due to the low yields obtained in preliminary procedures following the purification of catalase from pepper fruits, commercial catalase (527 aa) from bovine liver was used as model, as quite high protein concentrations are necessary to perform these *S*-nitrosation experiments. Enzyme activity was previously assayed in the presence of the *S*-nitrosation agent CSNO, which, when increased, caused a decrease in activity. (Figure 5a). This must be due to the NO component of CSNO, as incubation with cysteine alone did not affect bovine catalase activity (Figure 5b).

The procedure used to detect S-nitrosated cysteines uses an improved combination of nLC and MS to discriminate biotin-labeled polypeptides following trypsin digestion. Figure 6 shows the fragmentation spectrum (MS/MS) of peptides labeled with biotin-HPDP obtained after tryptic digestion of bovine catalase previously treated with CSNO. Two typical HPDP-biotinylated fragments are highlighted according to their corresponding signal intensities. Analysis of the mass/charge ratio of both fragments showed a 15 amino acid-long polypeptide, with an L_366_GPNYLQIPVNC_377_PYR_380_ sequence, in which biotin is putatively bound. This matches the reference sequence (NP_001030463.1) located between Leu366 and Arg380 obtained from the NCBI database. Thus, Cys377 from bovine catalase appears to be a candidate for *S*-nitrosation. The catalase aliquot, which was previously incubated with NEM, did not display any biotin labeling.

## 4. Discussion

### 4.1. Finely Tuned Modulation of Catalase Activity during Sweet Pepper Fruit Ripening

Catalase is one of the most important antioxidant enzymes in cells and a marker enzyme of peroxisomes. It is considered a redox guardian of these organelles due to its potentially important role in signaling processes in peroxisomes and other intracellular compartments [3]. Changes in catalase activity have been found to be a major monitoring index of plant responses under abiotic or biotic conditions [2,6,45,57,58]. A more recent study of expression profiling of both *A. thaliana* and *Oryza sativa* using transcriptomic and proteomic techniques, which erroneously reported that the catalase protein was also localized in the cytosol, found that catalase plays specific roles in development and stress responses [59].

We transcriptionally analyzed differential expression to evaluate changes in genes/proteins during the ripening process [56] and found that catalase activity decreased during the transition from the green to the red ripe fruit stage. Early results from the study of purified peroxisomes from both green and red pepper fruits show that catalase activity also decreased during maturation [35]. These results match those of previous studies that, using an antibody against the *A. thaliana* protein, showed a lower signal in red fruits, indicating that proteins can be regulated, at least at the translational level [18].

In addition, our findings indicate that catalase is finely regulated at the post-translational stage due to its susceptibility to several RNS-derived PTMs. Given the decrease in catalase activity in the treatment assays with SIN-1, DEA-NONOate, and GSNO (Figure 3), the enzyme appears to be liable to nitration and *S*-nitrosation (also called *S*-nitrosylation). SIN-1 triggers peroxynitrite formation by simultaneously generating both superoxide and nitric oxide radicals, which rapidly react to generate ONOO^−^, leading to the nitration of tyrosine residues in proteins [60]. On the other hand, both DEA-NONOate and GSNO are NO donors, which promote the *S*-nitrosation of cysteine residues. Furthermore, to determine whether lower enzyme activity in the presence of GSNO was due to NO or reduced glutathione (GSH) equivalents, assays were also carried out in the presence of GSH. We showed that catalase appears to be *S*-glutathionylated, indicating that GSNO, a physiological reservoir of NO capable of mediating transnitrosylation processes, may have a dual effect [54]. Our findings on the nitration of catalase from pepper fruit confirm a previous study that found this enzyme to be the most important nitrated protein [18]. As nitration and *S*-nitrosation events take place during pepper fruit ripening [18,38,61], catalase could be one of the main targets of these PTMs during this physiological process in which peroxisomes play a major role given their housing of the β-oxidation pathway and concomitant H_2_O_2_ generation.

Although the inhibitory effect of catalase activity caused by RNS was previously reported in plants, in some cases, *S*-nitrosation can activate other enzymes [62,63,64,65,66,67,68]. These NO-mediated post-translational modifications of catalase activity are important at the physiological level given that all these RNS signaling molecules are endogenously generated in peroxisomes, which are involved in important processes, such as fruit ripening, that contain other cell loci [35,40,69]. More recently, a new signaling gasottransmitter, hydrogen sulfide (H_2_S), which can induce a novel PTM called persulfidation, was shown to have an inhibitory effect on catalase activity [34].

### 4.2. Sweet Pepper Fruit Contains an Atypical Catalase Enzyme

Our assays carried out in the presence of increasing concentrations of H_2_O_2_ showed that catalase is susceptible to oxidation in the plant material (Figure 2). H_2_O_2_ is a commonly used substrate for catalase enzyme reactions, although the K_M_ of the enzyme is higher than that of other H_2_O_2_-scavenging peroxidases, such as ascorbate peroxidase [3]. However, the incubation of pepper samples in the presence of H_2_O_2_ levels of over 5 mM significantly inhibited catalase activity and altered electrophoretic mobility (Figure 2). This alteration in mobility, which was previously reported to be caused by oxidized SOD isozymes [70] and glucose-6-phosohate dehydrogenases [71], could also be due to other protein PTMs [72]. Protein oxidation usually involves the carbonylation of residues, which alters the molecular and kinetic properties of enzymes [73]. Nevertheless, castor bean endosperm catalase activity has been found to increase after treatment with H_2_O_2_. The effects of a diverse range of pro-oxidant situations, which either stimulate or depress catalase enzyme activity, have been reported [74,75,76,77,78]. The low concentrations of H_2_O_2_ used to inhibit the pepper fruit enzyme are somewhat atypical, as spectrophotometric determination of catalase commonly requires higher levels of H_2_O_2_ [3]. A possible inactivation of the enzyme by incubation conditions can be ruled out, as the real time stability assay indicated that catalase activity remained stable for at least 72 h (Figure 1).

The native molecular mass estimated for pepper fruit catalase is also atypical. The data obtained using two different approaches, non-denaturing electrophoresis and gel filtration, and those reported previously, which showed a monomer weight of 56 kDa [18,35], indicate that the protein is a homodimer. On the other hand, the native molecular mass for most plant catalases has been reported to be in the 220,000 to 240,000 Da range, with subunits of the protein homotetramer measuring approximately 55 to 59 kDa [1,3,4,79]. Another exception to this canonical structure was found in leaves from the halophyte *Mesembryanthemun crystallynum,* whose native catalase measured 320 kDa and the protein was composed of two dimeric forms each measuring 160 kDa. After denaturing PAGE, these dimers were further resolved into three 79-, 74-, and 62-kDa subunits [13].

Plant catalases present multiple isoforms encoded by a gene family, while the number and expression of the different isoforms vary throughout the plant’s development and according to environmental conditions [3,69]. Thus, in sunflower (*Helianthus annuus*) cotyledons, up to eight isozymes have been reported during the transition from glyoxysomes to leaf peroxisomes [80,81], while five isozymes were detected in cotton (*Gossypium hirsutum*) [82,83]. In *A. thaliana,* six isoforms encoded by three genes (*CAT1, CAT2*, and *CAT3*) have been found [84], while isoelectrofocusing analysis was used to differentiate five isozymes in purified peroxisomes from pea leaves [1]. A study of fruits from *Ziziphus mauritania* Lamk. (also known as Chinese date, jujube, and Indian plum) showed the presence of two catalase isozymes during the ripening process, only one of which was found at the initial immature green stage [85]. In pepper fruits, with a much higher catalase isoelectric point than that reported in plants, only one catalase isozyme was detected during the two ripening periods assayed [1,9].

### 4.3. Bovine Catalase (527 aa) is S-Nitrosated at Cys377

Due to pepper fruit sample limitations, catalase from bovine liver was used to examine the enzyme’s susceptibility to *S*-nitrosation. The protein sequence of bovine catalase contains 527 amino acids [86], and, using biotin switch labeling, we identified Cys377, located in the last third of the C-terminus, as a putative *S*-nitrosation target. To our knowledge, this is the first report of a Cys residue from catalase that can be post-translationally modified by *S*-nitrosation. This makes it particularly important to find target points where the enzyme can be modulated in other species under a diverse range of both physiological and adverse conditions. Catalase from *C. annuum*, which is reported to be composed of 492 amino acids [87], shows a 41% similarity to the bovine protein. Furthermore, 9 out of the 15 amino acids in the biotin-labeled polypeptide identified in bovine catalase coincide with a homolog fragment detected in pepper catalase, and the cysteine closest to this fragment is located at position 370. These dissimilarities indicate that more research is required to clearly establish where the protein can be modulated by this PTM in pepper fruit. Experiments on catalase gene overexpression in pepper fruit and additional purification, as well as in silico docking analyses would help to resolve these issues.

## 5. Conclusions

Due to its economic and nutritional repercussions, the key physiological process of fruit ripening, during which enormous metabolic changes occur in a genetically controlled environment, has become increasingly important in the field of crop research. This involves up- and down-regulation events that need to be analyzed with the aid of massive high-throughput techniques, such as RNA-Seq [56]. It is also necessary to study the most prominent proteins, particularly those associated with oxidative metabolism, and how they are affected by changing conditions during the ripening process, where the main ROS-related organelles, chloroplast, mitochondria, and peroxisomes, are involved. In this study, we characterized catalase, its potential modulation in pepper fruits, and its involvement in the ripening process. Catalase protein activity, which decreased during pepper fruit ripening, was also affected by ROS and RNS. This finely tuned dual regulation at both the translational and post-translational level suggests that the catalase protein could be required to maintain basic peroxisome cellular functions depending on the physiological stage of pepper fruits. This needs to be highlighted, as catalase is a moderately abundant protein exclusively located in peroxisomes, which are the major cellular source of H_2_O_2_ in plants [3,88,89]. In addition, catalase appears to play a central role in signaling processes driven by H_2_O_2_ during fruit ripening, in which ROS, RNS, and, more recently, H_2_S are clearly involved [18,34,35,38,60,90]. This role needs to be further investigated within the framework of the atypical molecular features manifested by the catalase enzyme in pepper fruits. Overall, the data reported here on the potential modulation of catalase by RNS and ROS and those from the *S*-nitrosation of the bovine enzyme could be translated to other biological systems, where catalases play important roles in the development and stress responses. This might be of particular interest in modulated processes in humans and also to address future research with catalase being the aim of structural, functional, and regulation analysis in order to understand the biological consequences of chronic exposure of cells to hydrogen peroxide leading to cellular adaptation [91]. Catalase is also a key enzyme in the ROS metabolism whose expression and localization are notably altered in tumors and some pathologies [91,92,93]. So, the data presented here on a specific modulation point by *S*-nitration could be an exploitable approach for upcoming research focused to better understand the role of catalase and its imbrication within the mechanisms underlying those unfavorable conditions.

## Figures and Tables

**Figure 1 antioxidants-08-00374-f001:**
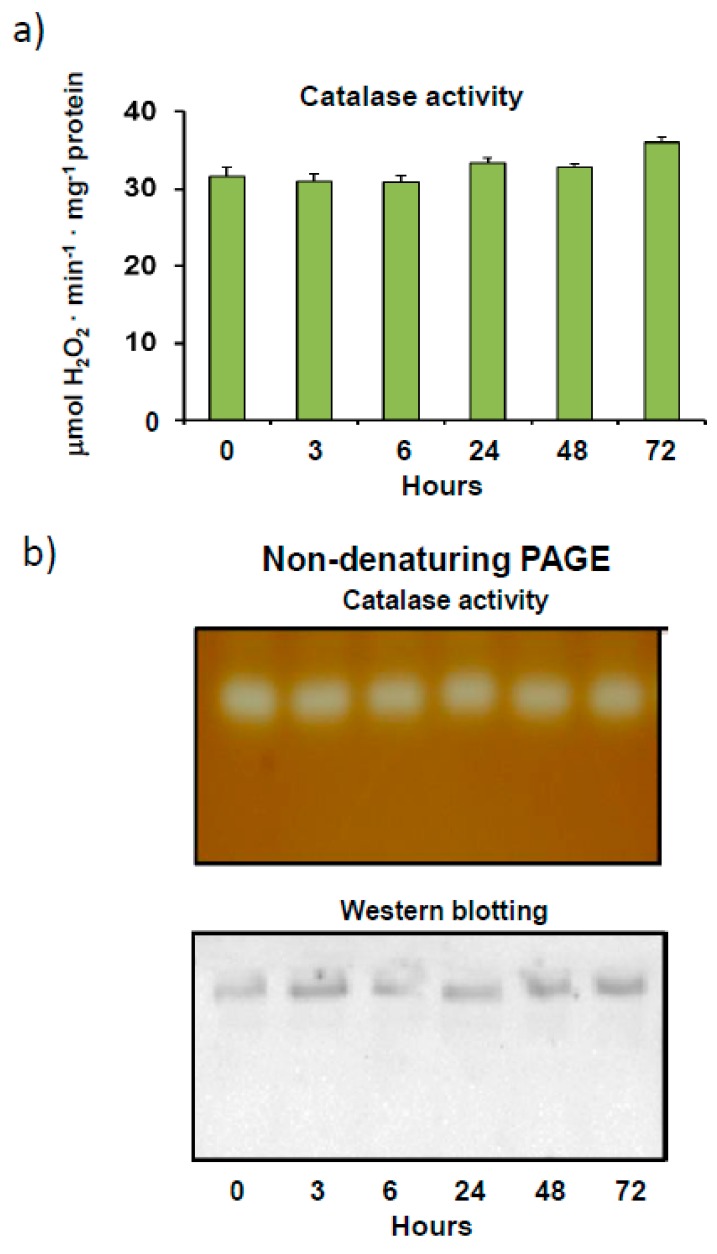
Catalase activity and protein content as a function of time in crude extracts from sweet pepper fruits. (**a**) Total enzyme activity. Samples were incubated at 4 °C for 0 to 72 h and then analyzed for catalase activity. Results are means ± SEM of samples from at least three different experiments, while the ANOVA test (*p* < 0.05) was used for statistical analysis. (**b**) Activity and protein level of the catalase isoenzyme as a function of time in pepper fruit. Prior to electrophoresis, samples were incubated at 4 °C for 0 to 72 h. Native electrophoresis was performed in 6% acrylamide gels for these assays. Green pepper fruit samples (20 μg protein) were used for catalase activity staining in gels. For western blot assays, pepper fruit samples (20 μg protein) were loaded onto gels and catalase was detected using an antibody against *Arabidopsis thaliana* catalase (1:5000 dilution). Shown pictures are representative of the non-denaturing PAGE and western blotting assays from at least three independent experiments.

**Figure 2 antioxidants-08-00374-f002:**
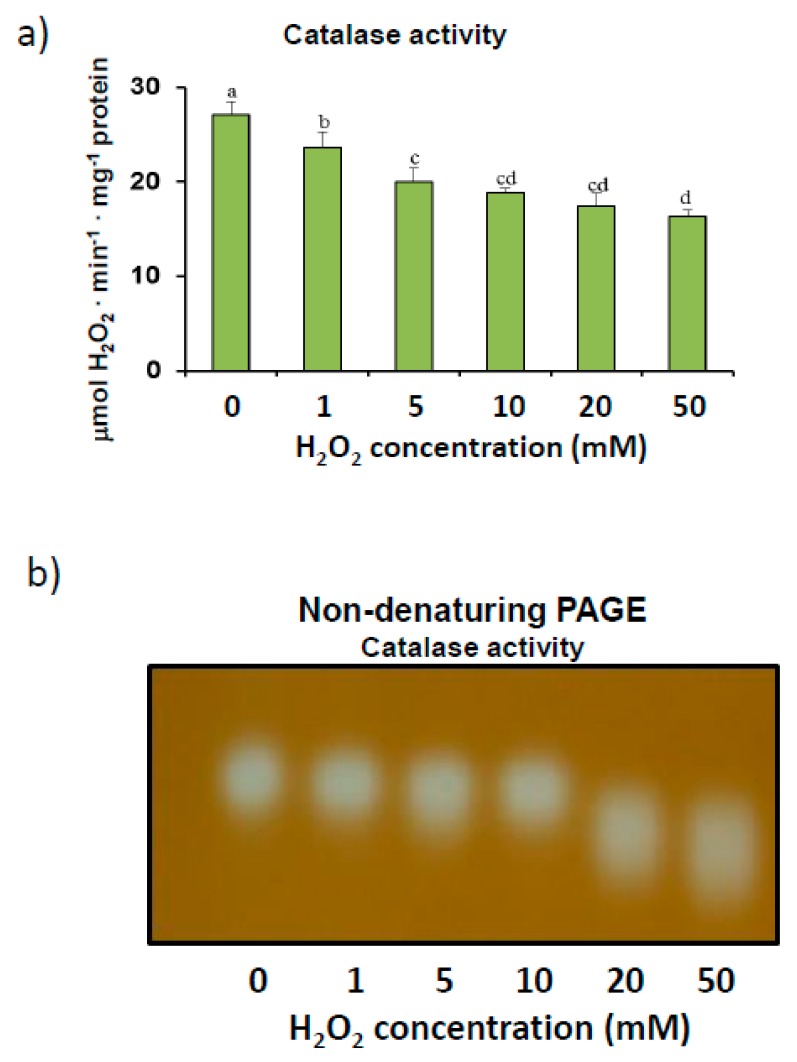
H_2_O_2_ modulation of catalase activity in sweet pepper fruits. (**a**) Total activity. Crude extracts from green peppers were incubated in the presence of 0 to 50 mM H_2_O_2_ for 30 min to determine catalase activity. Results are means ± SEM of samples from at least three different experiments. Columns with different letters are statistically distinct (ANOVA, *p* < 0.05). (**b**) Native PAGE (6% acrylamide) and catalase activity staining. Pepper fruit samples (20 μg) were treated with H_2_O_2_ (0–50 mM) for 30 min and then loaded onto gels. The shown picture is representative of the non-denaturing assays from at least three independent experiments.

**Figure 3 antioxidants-08-00374-f003:**
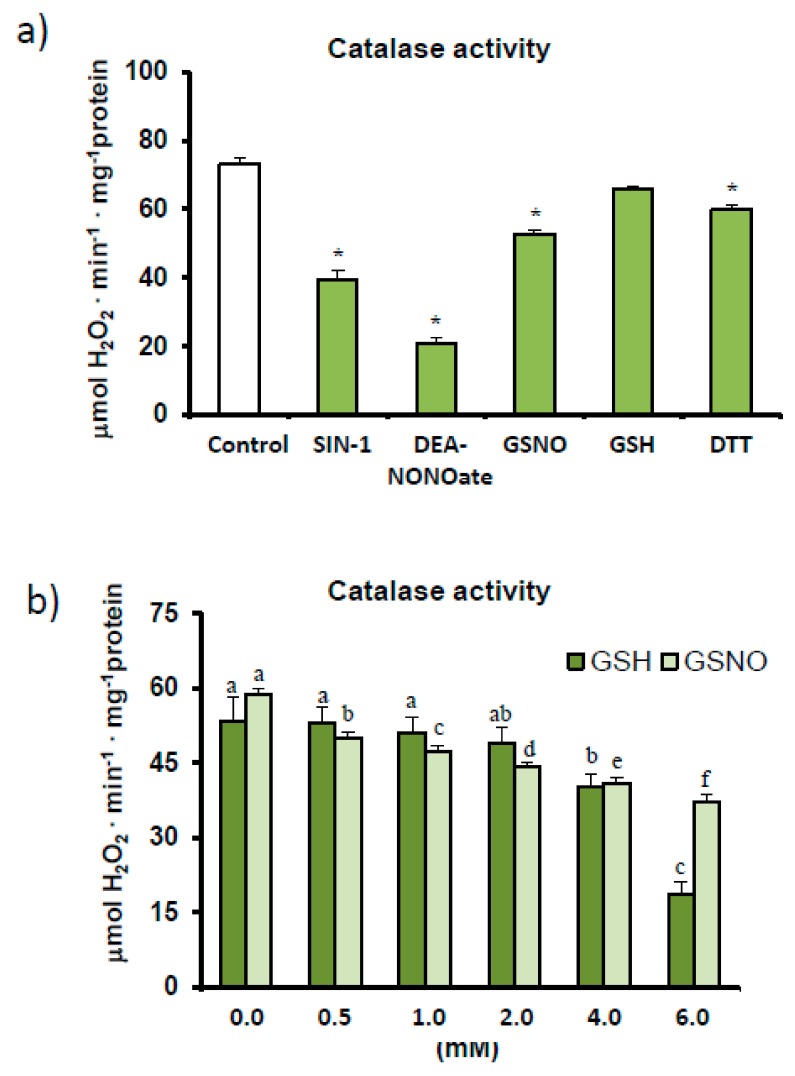
Effect of different modulating agents (ONOO^−^, NO, and reducing agents) on catalase activity in sweet pepper fruits. (**a**) Pepper samples were incubated for 45 min in the absence/presence of 2 mM 3-morpholinosydnonimine (SIN-1), 2 mM *S*-nitrosoglutathione (GSNO), 2 mM diethylamine NONOate (DEA-NONOate), 2 mM reduced glutathione (GSH), and 5 mM dithiotreitol (DTT), and catalase activity was then determined. Asterisks (*) denote significant differences in treatments with respect to control conditions in the absence of agents (student *t*-test, *p* < 0.05). (**b**) Effect of reduced glutathione and nitrosoglutathione on catalase activity in green fruits. Crude extracts were incubated with different concentrations of GSNO and GSH (0–6 mM) for 45 min (25 °C, darkness). Catalase activity was then determined spectrophotometrically. Results are means ± SEM of samples from at least three different experiments. Columns with different letters above each agent, either GSH or GSNO, differed statistically (ANOVA, *p* < 0.05).

**Figure 4 antioxidants-08-00374-f004:**
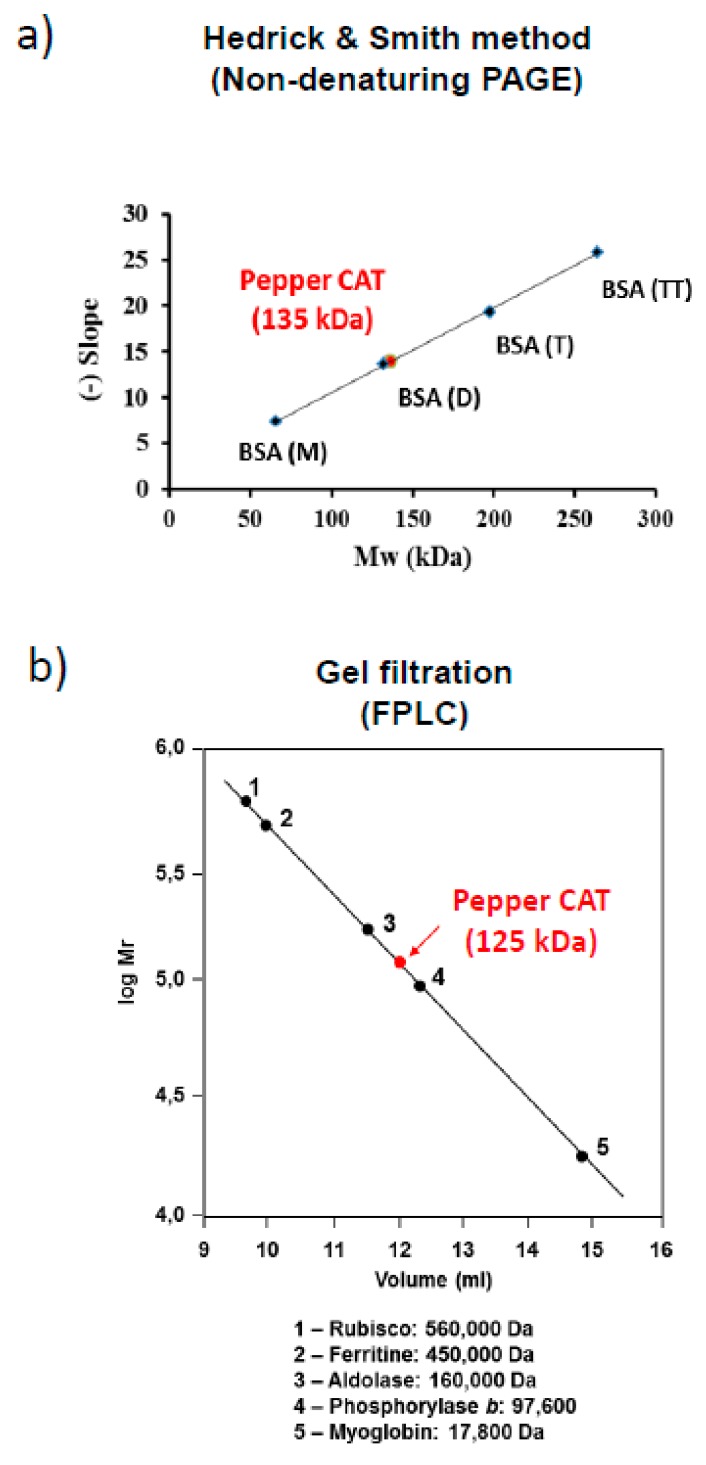
Estimation of native molecular mass of catalase in sweet pepper fruits. (**a**) Hedrick and Smith’s electrophoresis method. The slope was obtained by monitoring the electrophoretic mobility (Rf) of standard proteins on native gels and relative electrophoretic mobility (Rf) of pepper catalase at different acrylamide concentrations (5–7%). Monomeric, dimeric, trimeric, and tetrameric forms of bovine serum albumin were used as standards (BSA(M), BSA(D), BSA(T), BSA(TT)). (**b**) Fast Protein Liquid Chromatography (FPLC) gel filtration chromatography with a Superose 12 column was calibrated using the molecular mass of standard proteins, rubisco, ferritin, aldolase, phosphorylase b, and myoglobin.

**Figure 5 antioxidants-08-00374-f005:**
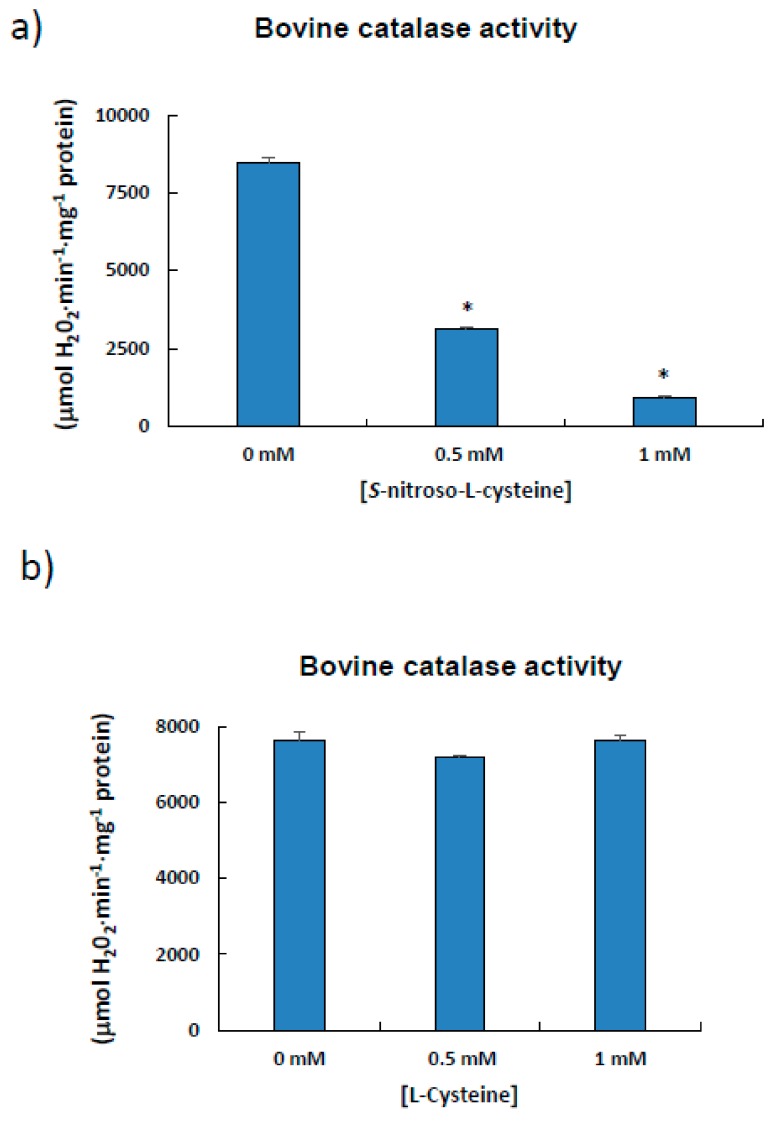
Effect of *S*-nitroso-L-cysteine (CSNO) on catalase from bovine liver. The enzyme (1 μL; Sigma-Aldrich, aqueous suspension, 10,000–40,000 units/mg protein) was incubated for 1 h at 25 °C in the absence/presence of different concentrations of either *S*-nitroso-L-cysteine (**a**) or L-cysteine (**b**) prior to determining catalase activity. Results are means ± SEM of samples from at least three different experiments. Asterisks (*) denote significant differences in treatments with respect to control conditions in the absence of agents (student *t*-test, *p* < 0.05).

**Figure 6 antioxidants-08-00374-f006:**
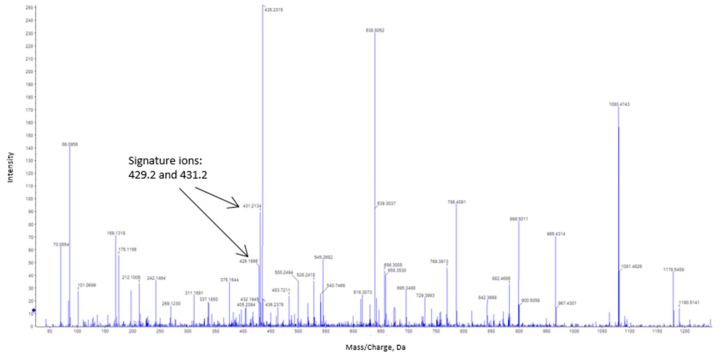
Tandem mass spectrometry (MSMS) fragmentation of peptides labeled with N-[6-(biotinamido)hexyl]-3′-(2′-pyridyldithio)propionamide (biotin-HPDP) following nanoliquid chromatography (nLC)-MSMS analysis of in-gel trypsin digestion of bovine catalase. The typical signature ions obtained from biotin-labeled catalase peptide are indicated by arrows. The amino acid sequence of the labeled peptide is L_366_GPNYLQIPVNC_377_PYR_380_.

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
