# Peer review of "Sweet Pepper (*Capsicum annuum* L.) Fruits Contain an Atypical Peroxisomal Catalase That Is Modulated by Reactive Oxygen and Nitrogen Species"

_antioxidants, 2019, doi:10.3390/antiox8090374_

Round 1

Reviewer 1 Report

The manuscript describes a molecular characterization of an atypical peroxisomal catalase from sweet pepper and its post-translational regulation by reactive oxygen and nitrogen species. It is a nice piece of scientific report, with a series of well-designed experiments, which are in general clearly described, the results presented and properly discussed. It brings highly interesting findings which represent important advances in the studies of the role of plant catalase in stress conditions associated with oxidative and nitrosative conditions.  However, there are several issues which need to be addressed to improve the clarity of the method or result description or discussion, as specified in detail below.

Major points

1.On my opinion, the first step of the used biotin switch procedure is not correctly described. In presence of N-ethylmaleimide, as described in L197, protein cysteine thiols would be rapidly and irreversibly modified by this reagent and thus not available as free thiols for S-nitrosylation modification by S-nitrosocysteine. Moreover, the use of NEM is not mentioned at all in the cited paper of Aroca et al. 2015. Authors should clarify this issue.

2.According to the literature, chemical reaction of reduced glutathione (GSH) and reduced protein cystein thiols cannot proceed to provide reversible protein S-glutathionylation. In contrast, GSH is able to react with “activated” thiol groups, ie. when the hydrogen atom is replaced by a –OH or –NO group. For this reason, S-glutathionylation should not be suggested as a possible mechanism explaining observed catalase inhibition by GSH. On the other hand, the inhibition observed by enzyme incubation with GSNO could be caused both by nitrosation or glutathionylation.

3.The description of results obtained by nanoLC-MS analysis presented in Fig.6 in the Result section is little bit confusing. The authors describe that “Fig 6 shows fragmentation spectrum (MS/MS) of bovine catalase” (L282-283). I would say that this Figure present rather a fragmentation spectrum of one of the peptides obtained by tryptic digestion of the catalase protein.

4.In the discussion, the authors discuss the previous findings of Alam and Ghosh (2018) as that “erroneously reported that the catalase protein is mainly localized in the cytosol” – this criticism of catalase cytosolic localization should be supported by appropriate scientific arguments.

Minor corrections

Introduction

-There is apparently a duplicated paragraph: the text between L62-72 is almost exactly duplicated in the next paragraph, i.e. L73-82, this need to be corrected

L48 correct plural form: “Catalases have highly divergent structures…”     

L59 “four isozymes having been identified” – correct to “four isozymes identified”

L65-66 correct to “… accumulated reports in recent years have suggested …

Methods:

L106 I believe the correct description should be “Frozen samples of pepper fruits were homogenized in liquid nitrogen and then extracted in …”

L118 authors should specify the activity of used HR peroxidase from Sigma-Aldrich, as this company offers multiple products with highly variable values of peroxidase specific activity; the same comment applies also to catalase from bovine liver  - the specific activity of the used Sigma-Aldrich product should be specified in L130-131

L123-128 the used concentrations of chemical compounds (SIN-1 etc.) should be included

L137 correct to “Pepper fruit samples were mixed with 0.006% (w/v) bromophenol blue dye“ – bromophenol blue is not a staining reagent, it is just an indicator of electrophoretic movement of the smallest compounds in PAGE gels

L154 I suggest to use “silver staining” instead of AgNO3

L161-162 It seems authors omitted to specify the used secondary antibody – please provide the type, company and used dilution.

L163 I suggest to change the heading to “2.6. Determination of molecular weight of native catalase enzyme” and then to use “molecular weight” instead of molecular size throughout the manuscript

L197 correct “N-ethylmaleimide”

Results

L226: the meaning of this part of the phrase is not clear: “(catalase)… is present in down-expressed proteins” – this should be rewritten

Legend of Fig.1: correct “Catalase isoenzymatic activity and protein expression …” to “Catalase activity and protein level …; correct “pepper fruit samples (20 μg protein) were loaded onto gels with an antibody against A. thaliana catalase ..” to “pepper fruit samples (20 μg protein) were loaded onto gels and catalase was detected using with an antibody against A. thaliana catalase ..”

L257 correct S-nitrosation

Discussion

L373 correct the phrase “… hydrogen sulfide (H2S), which could undergo a novel PTM called persulfidation” to “…. hydrogen sulfide (H2S), which can induce a novel protein PTM called persulfidation”

Reviewer 2 Report

The manuscript by Rodriguez-Ruiz et al. describes the molecular and biochemical characterization of catalase in pepper fruits and its post-translational modification by ROS/RNS species. The authors found that catalase activity decreased during the fruit ripening process. Notably, the catalase from bovine liver demonstrated s-nitrosation at Cys377. Overall this study provides evidence for the presence of an atypical catalase enzyme in pepper fruits that can be modulated by NO-derived molecules and reducing agents.

In general, experiments are well designed and carefully performed. The data appeared to be reliable and well-supported conclusions.

Specific points:

1) Include loading control data for the western blot result. The authors could probe with loading control antibodies or perform a simple Ponceau staining.

2) Authors should mention how many biological/technical replicates were used for catalase activity assays and to perform statistical tests. It’s unclear if the data shown [Fig. 1 (a), 2 (a), 3 (a, b), and 5 (a, b)] is average of #n replicates or values are shown from one experimental repetition? Please clarify.

Reviewer 3 Report

The manuscript By Rodríguez-Ruiz et al. describes the molecular mechanisms responsible for the modulation of peroxisomal catalase activity in the Sweet Pepper fruits. In particular, the authors demonstrate the existence of an atypical peroxisomal catalase, which is modulated by reactive oxygen and nitrogen species. The study is interesting in its topic and only partially innovative since the authors identify a new cysteine involved in the post translational modification of catalase and thus related to the changes in enzyme activity occurring during the ripening of sweet pepper fruits, in addition to the others already identified aminoacids.  These findings would provide additional information to the fine regulation of catalase activity. The experiments are well conducted and the conclusions partially supported by the results obtained. To this reason, further experiments should be performed to strengthen the working hypothesis.

Major questions:

1.         Although catalase is the most abundant enzyme in Sweet pepper fruits, the expression and activity of other antioxidant enzymes such as SOD and Glutathione peroxidase should be measured.

2.         The endogenous production of NO should be assessed as well as the NOS expression and activity, in the different step of maturation of Sweet pepper fruits. These experiments will help to better understand the effect of NO-donors on the modulation of catalase activity.

3.         Finally, the contribute of mitochondria to the regulation of oxidative metabolism during the ripening of sweet pepper fruits might be helpful to discriminate the source of ROS and RNS production into the cells in this phenomenon. 

4.         In the discussion, the authors should more emphasize the relevance and the translability of their novel results.

Round 2

Reviewer 3 Report

The authors did not perform the experiments required but provided their references in support of the referee's suggestions. In some of the references cited by the authors are reported indirect evidences of the experiments required by the referee, in particular for NO measurements. This strategy could be accepted but the study loses in novelty. Indeed, the information provided in the present study are confirmative of the previous studies adding just a small a detail related to the identification of the cysteine that, in the catalase, might be responsible for the atypical functional properties. 
Moreover, the translability of this achievement is still not adequately emphasized
